# Phenotypic Variation in Clinical *S. aureus* Isolates Did Not Affect Disinfection Efficacy Using Short-Term UV-C Radiation

**DOI:** 10.3390/microorganisms11051332

**Published:** 2023-05-18

**Authors:** Birte Knobling, Gefion Franke, Laura Carlsen, Cristina Belmar Campos, Henning Büttner, Eva M. Klupp, Philipp Maximilian Maurer, Johannes K. Knobloch

**Affiliations:** Department Infection Prevention and Control, Institute for Medical Microbiology, Virology and Hygiene, University Medical Center Hamburg-Eppendorf, Martinistraße 52, 20246 Hamburg, Germany; b.knobling@uke.de (B.K.); la.carlsen@uke.de (L.C.); p.maurer@uke.de (P.M.M.)

**Keywords:** pigmentation, catalase activity, biofilm, UV-C, disinfection, tolerance

## Abstract

Pigmentation, catalase activity and biofilm formation are virulence factors that cause resistance of *Staphylococcus aureus* to environmental stress factors including disinfectants. In recent years, automatic UV-C room disinfection gained greater importance in enhanced disinfection procedures to improve disinfection success in hospitals. In this study, we evaluated the effect of naturally occurring variations in the expression of virulence factors in clinical *S. aureus* isolates on tolerance against UV-C radiation. Quantification of staphyloxanthin expression, catalase activity and biofilm formation for nine genetically different clinical *S. aureus* isolates as well as reference strain *S. aureus* ATCC 6538 were performed using methanol extraction, a visual approach assay and a biofilm assay, respectively. Log_10_ reduction values (LRV) were determined after irradiation of artificially contaminated ceramic tiles with 50 and 22 mJ/cm^2^ UV-C using a commercial UV-C disinfection robot. A wide variety of virulence factor expression was observed, indicating differential regulation of global regulatory networks. However, no direct correlation with the strength of expression with UV-C tolerance was observed for either staphyloxanthin expression, catalase activity or biofilm formation. All isolates were effectively reduced with LRVs of 4.75 to 5.94. UV-C disinfection seems therefore effective against a wide spectrum of *S. aureus* strains independent of occurring variations in the expression of the investigated virulence factors. Due to only minor differences, the results of frequently used reference strains seem to be representative also for clinical isolates in *S. aureus*.

## 1. Introduction

Multidrug resistance has increased worldwide and is considered a threat to public health. Several recent investigations have reported the emergence of multidrug-resistant (MDR) bacterial pathogens from different origins, which increase the necessity of the proper use of antibiotics and effective disinfectants [1,2]. Routine susceptibility testing against antibiotics as well as disinfectants of emerging MDR strains is required to optimize infection prevention.

Methicillin-resistant *Staphylococcus aureus* (MRSA), defined as multidrug-resistant [3], is still one of the common pathogens causing healthcare-acquired infection, although the overall proportion of methicillin resistance within clinical *S. aureus* isolates has decreased in recent years [4]. In addition to the classical healthcare-associated MRSA (HA-MRSA), clones that spread in the community (CA-MRSA) or are acquired by contact with livestock (LA-MRSA) have also emerged outside hospitals in the past [5]. As a colonizer, *S. aureus* is part of the normal human skin flora, but it can also cause a variety of infectious diseases, ranging from mild skin infections to deadly infections such as pneumonia and sepsis. Antibiotic therapy of *S. aureus* infections can, besides methicillin resistance, be affected by different resistance mechanisms. *S. aureus* can produce a particularly high number of different toxins and other virulence factors [6].

Among other aspects, the high tenacity with long survival time of *S. aureus* on inanimate surfaces is a relevant cause for frequent detection in the environment surrounding patients [7]. This has recently been demonstrated in studies investigating MRSA on surfaces in patient rooms occupied by colonized or infected patients, which found up to 40% [8,9] of surfaces contaminated. Furthermore, a recent investigation using fluorescent marker technology to verify compliance with cleaning and disinfection processes found that only 63% of surfaces were adequately cleaned [10]. In addition, measuring aerobic colony counts, Chen et al. determined 18.8% of surfaces to be contaminated by MRSA even after manual terminal disinfection of patient rooms [11]. Subsequent patients were at increased risk of acquiring nosocomial infection or new emerging colonization if a patient infected or colonized with MRSA or other environmentally stable pathogens had previously been cared for in the room [12,13]. The probable cause for this could be insufficient manual cleaning and disinfection.

Due to inadequate manual disinfection processes, an increasing number of different enhanced disinfection procedures have been established to improve room disinfection efficacy. One common autonomous disinfection procedure is radiation with high energetic short-wave ultraviolet light (UV-C). The high effectiveness of UV-C radiation in reducing a wide range of microorganisms, including MRSA, has been demonstrated, although predominantly in experimental studies [14,15]. As known for conventional chemical disinfectants, in vitro efficacy can vary greatly from real-world disinfection in a complex hospital setting depending on factors such as surface area, disinfectant concentration, contact time and type of microorganism [16]. Until now, mainly representative laboratory and test strains have been used for testing disinfection procedures. These standard test organisms used as reference strains may behave differently to disinfectants than the wide variety of bacterial clones found in the hospital environment. This fact is now taken into account in the European standard for testing chemical disinfectants by proposing experiments using naturally occurring pathogens under usual conditions of use [17].

It has been shown that damage to DNA, RNA and proteins is the main effect of inactivation of pathogens by UV-C radiation [18]. In defence against this and against treatment with antibiotics, *S. aureus* has various DNA repair mechanisms such as photoreactivation by DNA photolyase enzymes and mechanisms of detoxification [18]. Therefore, it can be assumed that these could influence the tolerance to UV-C. In addition, *S. aureus* is known to be equipped with various virulence factors that induce distinctive protective properties [19]. Golden pigment, contributed by the carotenoid staphyloxanthin [20], and catalase, which is a ubiquitous antioxidant enzyme [21], are two of those virulence factors produced by *S. aureus* that are associated with survival on inanimate surfaces. Additionally, the expression of extracellular polysaccharides acting as intercellular adhesins and mediating the capacity to form multi-layered biofilms on inanimate surfaces might add to protection against environmental stress factors.

In a comparison of wild-type *S. aureus* with strong staphyloxanthin expression and *crt*-mutants with severely reduced staphyloxanthin production, the mutant strain has been shown to be more susceptible to UV-C radiation [19]. This suggests that the production of staphyloxanthin and therefore pigmentation could have an influence on the tolerance of *S. aureus* to UV-C radiation.

The production of catalase has been shown to be a relevant protector against hydrogen peroxide, which is a common component of immune defence but also a relevant disinfectant in the hospital environment [21]. The protective properties of catalase were supported by a recent study showing increased resistance of *S. aureus* to vaporized hydrogen peroxide compared to the biological indicator organism *Geobacillus stearothermophilus* [22]. Pottage et al. suspected the catalase activity as a possible cause [22]. The damage of microorganisms by UV-C radiation is based, among other mechanisms, on reactive oxygen species produced during absorption [23]. Since catalase is able to degrade these, differences in the ability of clinical *S. aureus* isolates to produce catalase could lead to differences in UV-C tolerance.

It is known that bacterial biofilms survive more effectively on environmental surfaces and are more tolerant to disinfection than planktonic bacteria [24]. The production of extracellular polymeric substances enables the protection of bacteria within the biofilm from direct contact with disinfectants [24]. This shielding has also been reported in UV-C disinfection as a reason why biofilms displayed greater tolerance to UV-C than planktonic cells [25,26]. As it has been shown that biofilms on surfaces in the hospital environment frequently contain *S. aureus* [27], variations in the capacity of *S. aureus* to form biofilms could have an influence on the UV-C tolerance. Biofilm formation in staphylococci is mediated predominantly by the expression of the polysaccharide intercellular adhesin (PIA), also known as poly-N-acetyl-β-(1–6)-glucosamine (PNAG) in *S. aureus* [28], mediating cell-to-cell adhesion after primary attachment to inanimate surfaces [29]. PIA/PNAG is synthesized by the gene products of the *ica*ADBC gene cluster [29]. Interestingly, in contrast to *Staphylococcus epidermidis*, which frequently lacks this operon, in *S. aureus* the *ica*ADBC operon could be suggested as part of the core genome [30]. However, despite the presence of *ica*ADBC, not all *S. aureus* strains display a biofilm-positive phenotype under conditions in which *ica*ADBC-positive *S. epidermidis* strongly express PIA synthesis [30], indicating a differential regulation of the PIA/PNAG synthesis between these species [29].

The relative impact of catalase formation, pigmentation and biofilm formation on tolerance to automatic room disinfection procedures using UV-C radiation is of great relevance. Especially because there is no standardised test procedure for evaluating UV-C disinfection efficacy so far [31], this knowledge is essential in order to perform recommended implementation investigations with representative pathogens in the area of use [32].

Therefore, the aim of the study was to assess a possible connection between staphyloxanthin production, catalase expression and biofilm formation with the level of UV-C tolerance in *S. aureus*. For this, well-characterized genetically distinct clinical *S. aureus* isolates were compared to the standard test organism ATCC 6538 used for normative testing of disinfection procedures. UV-C susceptibility was measured under reality-simulating test conditions in order to be able to derive a conclusive value for the application in the hospital setting.

## 2. Materials and Methods

All *S. aureus* isolates used were identified in routine diagnostics using MALDI TOF mass spectrometry and the MALDI Biotyper^®^ database (Bruker, Bremen, Germany), and antimicrobial susceptibility testing was performed using a VITEK instrument (bioMérieux, Marcy l’Etoile, France). The species was confirmed, and strains were further characterised by whole genome sequencing with subsequent genomic analysis as described recently [33]. From a large strain collection, eight different clinical MRSA isolates including clones, which are described in the international literature (e.g., [5,34]) as HA-MRSA (ST5/t010, ST8/t024, ST22/t223, ST45/t015), CA-MRSA (ST30/t300, ST59/t216), and LA-MRSA (ST1/t127, ST398/t034), and a low pigmented clinical methicillin-susceptible *S. aureus* isolate (MSSA ST45/unknown spa-type) were selected. These clinical isolates, as well as the standard test organism *S. aureus* ATCC 6538 (ST464/t3297), were investigated under standardized conditions. All isolates harbored the *ica*ADBC gene cluster required for PIA/PNAG synthesis.

### 2.1. Quantification of Staphyloxanthin Expression

Quantification of *S. aureus* staphyloxanthin expression was investigated with adaptations based on a methanol extraction assay [20]. Briefly, *S. aureus* isolates were simultaneously grown aerobically on Tryptic Soy Agar (Oxoid Deutschland GmbH, Wesel, Germany) for 48 h at 37 °C. Using a sterile inoculation loop, bacteria were completely transferred to a 1.5 mL micro tube (Sarstedt AG & Co. KG, Nümbrecht, Germany) containing 1 mL sterile water (Merck KGaA, Darmstadt, Germany). Cells were harvested by centrifugation (Fresco^TM^ 21 Microcentrifuge, Thermo Fisher Scientific, Waltham, MA, USA) at 17,000× *g* for 3 min. Each cell pellet was washed twice in sterile water and finally resuspended in 420 µL methanol (Th. Geyer GmbH & Co. KG, Renningen, Germany). Then, 20 µL of this suspension were added to 980 µL sterile water in order to measure the optical density of each sample at 600 nm (OD_600_) using a Sparke^®^ Microplate reader (Tecan Trading AG, Männedorf, Switzerland) in quadruplicate with each 200 µL of the suspension in a 96-well plate. The remaining 400 µL suspensions were incubated for 5 min at 55 °C using a laboratory block heater (FastGene Mini Dry Bath, NIPPON Genetics EUROPE, Düren, Germany), followed by centrifugation for 2 min. For quantification of staphyloxanthin, 350 µL of the suspension were added to 650 µL methanol containing micro tube. Finally, absorbance at 465 nm (A_465_) was measured and normalized against the OD_600_ of the low pigmented *S. aureus* ST45 isolate. Three independent experiments in duplicate were conducted.

### 2.2. Measuring Catalase Activity

Catalase activity was quantified using a visual approach as described previously with adaptations [21]. In brief, for each MRSA or MSSA isolate as well as the reference strain, a suspension with an optical density of McFarland 2 was produced in a 0.9% NaCl solution. In each case, 1 mL of the suspension was transferred to a 10 mL tube, and then 100 µL of 1% Triton-X100 (Th. Geyer GmbH & Co. KG, Renningen, Germany) solution and 100 µL hydrogen peroxide 30% (Th. Geyer GmbH & Co. KG) were added. After thorough mixing, a reaction time of 15 min was allowed, and finally the height of the resulting visible foam rising in the test tube was measured using a ruler. The experiment was conducted independently three times.

### 2.3. Quantification of Biofilm Formation

The ability to form a biofilm under different growth conditions was determined with adaptations following Mack et al. [35] and Knobloch et al. [36]. In brief, 20 µL of pre-culture grown for 4 h in tryptic soy broth (TSB, Becton Dickinson GmbH, Heidelberg, Germany) were added to either 2 mL TSB, TSB supplemented with 4% NaCl (wt/vol) or 4% ethanol (vol/vol). Then, 200 µL of each suspension was added to eight wells of a 96-well plate and incubated at 37 °C for 20–24 h. Subsequently, each well was washed four times with Dulbecco’s phosphate-buffered saline (anprotec, Bruckberg, Germany). After drying, wells were stained with gentian violet solution (Carl Roth GmbH + Co. KG, Karlsruhe, Germany) for 15 min and dried again. Finally, the absorbance was measured with the Sparke^®^ Microplate Reader at 570 nm (A_570_) and the reference value at 405 nm. A_570_ values <0.1 were defined as biofilm-negative whereas values between 0.1 and 0.5, and values >0.5 corresponded to weak and strong biofilm formation, respectively. Values exceeding the limit of the instrument (A_570_ > 4.0) were set to 4.0 for calculation of mean values. In addition to the tested *S. aureus* isolates, biofilm-negative *S. epidermidis* 1585 and biofilm-positive *S. epidermidis* 1457 were carried along as negative and positive controls, respectively. For each growth condition, three independent experiments were performed in duplicate.

### 2.4. UV-C Disinfection Procedure

The setup of the UV-C disinfection procedure was defined according to the manufacturer’s instructions for the use of a mobile UV-C light source, UVD robot model C (UVD Robots, Odense S, Denmark), emitting a wavelength of 254 nm. The manufacturer specifies a minimum UV-C light intensity of 2500 µW/cm^2^ at a distance of 1 m. According to Equation (1), an exposure time of 20 s was defined to reach UV-C doses of approx. 50 mJ/cm^2^ at 1.0 m and 22 mJ/cm^2^ at 1.5 m distance. Since the microorganism-time-calculator of the manufacturer specifies a required UV-C dose of 104 J/m^2^ (approx. 10 mJ/cm^2^) to achieve Log 4 reduction, this setting should be effective to reduce *S. aureus*.
(1)$$\mathrm{UVC}\;\mathrm{dose}\;\left[\frac{\mathrm{mJ}}{{\mathrm{cm}}^{2}}\right]=\mathrm{intensity}\left[\frac{\mathrm{mW}}{{\mathrm{cm}}^{2}}\right]\times \;\mathrm{exposure}\;\mathrm{time}\;\left[s\right]$$

This calculation was confirmed in individual experiments checking the color change of UV-C indicators (UV-C Dosimeter^TM^, UVD Robots, Odense S, Denmark) after irradiation with these settings.

To avoid the influence of the three-minute warm-up phase of the UVD robot on the reduction rates, the warm-up phase took place in an anteroom. Subsequently, the robot automatically moved to the predefined disinfection point in the test room, which was located at the above-mentioned distance from the samples. After completing the irradiation, the robot switched off immediately at the disinfection position.

### 2.5. Testing UV-C Tolerance Using Reality-Simulating Experimental Conditions

The clinical *S. aureus* isolates as well as the reference strain were used to prepare standardized contaminated surfaces under clean conditions [17]. For each isolate, 20 µL of a bacterial suspension in 0.3% bovine serum albumin in 0.85% sodium chloride solution (range 1.6–4.4 × 10^8^ colony-forming units [cfu] per mL) was spread on ceramic tiles (5 × 5 cm, #3709PN00, Villeroy&Boch, Mettlach, Germany) using a sterile spatula (Sarstedt AG & Co. KG), and dried completely.

Subsequently, contaminated surfaces were positioned in duplicate on a one-meter height black table at 1.0 and 1.5 m distances. Furthermore, two contaminated surfaces per isolate were stored outside the test room as untreated controls. For each isolate, five independent experiments were performed. To account for the influence of variations in environmental conditions, the relative air humidity and temperature were recorded in each experiment via a LogTag Trex-8 (CiK Solutions GmbH, Karlsruhe, Germany).

To calculate reduction factors, treated surfaces and controls were wiped off in a standardized way with previously moistened flocked nylon swabs (eSwab^TM^ Standard, Copan; Brescia, Italy) in order to recover bacteria from ceramic tiles. Swabs were then added to the Amies medium and vortexed for 30 s to elute recovered bacteria; subsequently, 100 µL was spread in duplicate on Columbia agar containing 5% sheep blood (bioMérieux) to carry out quantitative culture (detection limit 5 cfu/25 cm^2^). Agar plates were incubated at 37 °C for 18–24 h. Finally log reduction values (LRV) were calculated. If there was no growth in both approaches, the reduction value was reported as log_10_ of the control according to DIN EN 17272:2020 [37]. Five independent experiments were conducted.

Statistical analysis was performed using R (version 4.0.3) [38] and R studio (version 2021.09.1) [39] with activated packages *ggpubr* [40] and *rstatix* [41]. To detect differences between the reduction factors of the clinical *S. aureus* isolates and standard test organism *S. aureus* ATCC 6538, one-way ANOVA was conducted. In case of significant results, a subsequent pairwise post-hoc Tukey test was performed. For all analyses, a confidence level of 95% was used. However, due to naturally occurring variations in microbial growth, only *p*-values <0.01 were assessed as statistically significant for pairwise comparisons.

## 3. Results

Staphylococci are commensals of the skin and therefore have protective factors that protect themselves from light exposure and free oxygen radicals. In this study, the sensitivity to a disinfection procedure using UV-C of representative MRSA isolates compared to a very low pigmented MSSA and the laboratory strain ATCC should be investigated. All investigated isolates were phenotypically characterized.

### 3.1. Genotypic and Phenotypic Characteristics of the Investigated Isolates

Genotypic and phenotypic characterisation confirmed a *mec*A gene as well as phenotypic resistance against oxacillin in all eight MRSA isolates. Additional phenotypic and genotypic resistance was observed in five MRSA isolates (Appendix A). However, none of the isolates were considered as extensively drug-resistant or (XDR) or pandrug-resistant (PDR) [3]. In MSSA isolates, no genotypic or phenotypic resistance was observed.

Staphyloxanthin quantification revealed median values of the normalized A_465_ between 0.325 and 0.109 (Figure 1A). *S. aureus* ATCC 6538 showed the highest absorption, representing the highest amount of synthesized staphyloxanthin (median 0.325, IQR 0.067), followed by MRSA ST58/t216 (median 0.245, IQR 0.095). The other MRSA isolates showed a lower variance of staphyloxanthin expression with A_465_ values in a range of 0.19 to 0.13. As expected, the methicillin-susceptible *S. aureus* (MSSA) ST45 that presented as atypical in routine clinical diagnostics and lacked the typical yellow staining showed the lowest A_465_ value (median 0.109, IQR 0.005). In general, the quantified staphyloxanthin expression was in accordance with the visual comparison of the investigated *S. aureus* isolates.

Quantification of the catalase activity revealed median values in a range of 1.4 to 7.2 cm foam height in the visual assessment (Figure 1B). The strongest catalase activity was observed in *S. aureus* ATCC 6538 (median 7.2, IQR 0.4). MRSA ST8/t034 (median 1.4, IQR 0.5) showed the lowest catalase activity. MSSA ST45, MRSA ST398, MRSA ST45, MRSA ST1 and MRSA ST30 showed comparable catalase activity in a lower medium range (median 2.7 to 4.1). Greater activity, but also in a medium range, was observed for MRSA ST59, ST22 and ST5 (median 5.2 to 6.2).

Analysis of the capacity to form biofilms under non-inducing conditions revealed two isolates expressing a weak (MRSA ST8/t024 and ST5/t010) and one isolate (MRSA ST45/t015) expressing a strong biofilm-positive phenotype (Figure 1C). Induction of biofilm formation by supplementing the growth medium with NaCl was observed in five isolates (Figure 1C). Thereby, three isolates with a biofilm-negative phenotype under uninduced conditions displayed a weak (MRSA ST22/t223) or strong (MRSA ST1/t127 and ST59/t216) biofilm-positive phenotype. The biofilm-positive MRSA isolates ST45/t015 and ST8/t024 showed further increased biofilm formation induced by NaCl supplementation. In contrast, weak biofilm-forming MRSA ST5/t010 showed the lowest A_570_ in NaCl-supplemented medium.

Supplementation of the medium with sub-inhibitory concentrations of ethanol resulted in an induction of biofilm formation in strain MRSA ST45/t015 but reduced A_570_ values in strains MRSA ST8/t024 and ST5/t010 with a still weak and biofilm-negative phenotype, respectively. No positive biofilm phenotype was observed in four isolates under any of the conditions tested (*S. aureus* ATCC 6538 and MSSA ST45 as well as MRSA ST30/t300 and MRSA ST398/t0349). A strong biofilm formation could be detected under all test conditions for MRSA ST45/t015.

### 3.2. UV-C Disinfection of Investigated Isolates

For all isolates tested, the average bacterial concentration on the contaminated surfaces ranged from 8.7 × 10^5^–4.5 × 10^6^ cfu/25 cm^2^ (Appendix A). Comparable room air conditions were observed for all experiments. The log reduction value (LRV) after irradiation with a UV-C dose of 50 mJ/cm^2^ ranged between median 4.75 to 5.94 (Figure 2). Therefore, the maximum difference of LRV was 1.19. The largest reduction was observed for MRSA ST30/t300 (median 5.99) and *S. aureus* ATCC 6538 (median: 5.94). MRSA ST5/t010 (median: 4.86) and MRSA ST22/t223 (median: 5.07) were observed to be the most tolerant isolates.

Irradiation with the lower UV-C dose of 22 mJ/cm^2^ resulted in an LRV between median 4.68 and 5.11 (Figure 2). The difference of LRV is at maximum 0.43 between the most susceptible and the most tolerant isolate.

To determine statistically significant differences in LRV, one-way ANOVA was performed. A significant result was shown for the UV-C dose 50 mJ/cm^2^ (F(9, 90) = 3.66, *p* = 6.18 × 10^−4^, generalized eta squared = 0.268). Pairwise comparisons resulted in a significant *p*-value of 0.00056 for the comparison MRSA ST5/t010 to MRSA ST30/t300 (CI: −2.03 to −0.350). The further comparisons did not show significant *p*-values as defined. When performing a one-way ANOVA to compare the LRV determined with a UV-C dose of 22 mJ/cm^2^, no significant difference was found.

## 4. Discussion

The disinfecting effect of UV-C radiation is attributed to the absorption of energy by proteins and nucleic acids such as DNA, promoting the development of molecular rearrangements and photoproducts as well as the formation of reactive oxygen species [18]. *Staphylococcus aureus* strains are equipped with a variety of mechanisms to defend against oxidative stress caused by exogenous and endogenous reactions [42]. Pigmentation, mainly based on the pigment staphyloxanthin, is known as a protective factor against desiccation, photosensitization and reactive oxygen species [42]. Further protective factors against oxidative stress in staphylococci are the enzymes catalase and superoxide dismutase. A complex regulatory network modulates the expression of protective factors of staphylococci against oxidative stress and other damaging environmental conditions [43]. This network is balanced differently in individual strains even without the presence of environmental stress factors, so that the expression levels of these protective factors often differs between individual strains [42,43]. In addition to the factors mentioned, the ability to form biofilms is generally known to increase the environmental resistance of bacteria. Staphylococcal biofilm formation has been reported to be induced by environmental factors and stress conditions [44]. *S. aureus* has also been observed with high prevalence in dry biofilms on surfaces in the hospital environment [27]. This suggests that biofilm formation is also a common protective factor that increases the survival of *S. aureus* and tolerance to disinfectants.

When exposed to fast-acting disinfectants such as short-term irradiation with high-energy UV-C radiation, microorganisms cannot react by adapting the expression of protective factors. Therefore, the investigation of different isolates with natural differences in the expression of preformed protective factors is of high importance for the assessment of the effectiveness of disinfection measures in the hospital setting.

By comparison of eight different clinical MRSA isolates of different clonal complexes, strong variations in pigment expression were observed. The normative test organism ATCC 6538 showed conspicuously strong pigment in contrast to *S. aureus* ST45, which had the lowest pigmentation. Differences were also found in catalase activity, with ATCC 6538 *S. aureus* displaying the highest catalase activity but no association between catalase activity and pigmentation for the other isolates.

Six of the investigated *S. aureus* strains displayed the capacity to form a biofilm under at least one of three different culture conditions, whereas four isolates displayed a biofilm-negative phenotype under all tested conditions. Only MRSA ST45/t015 displayed a strong biofilm-positive phenotype under all tested growth conditions, indicating that bacterial clusters embedded in PIA/PNAG could be present on the contaminated surfaces used for the evaluation of the UV-C disinfection process. More importantly, despite the presence of the *ica*ADBC operon in all strains, the different biofilm phenotypes of the individual strains confirm that strong differences in the global regulatory networks of *S. aureus* are present in these strains and could influence the susceptibility against UV-C radiation.

The log reduction value (LRV) achieved at a UV-C radiation dose of 50 mJ/cm^2^ was greater than five log_10_ for all isolates tested with the exception of MRSA ST5/t010. The observed differences in LRV were statistically significant (*p* < 0.01) only when comparing MRSA ST5/t010 and MRSA ST30/t300. MRSA ST5/t010 was identified as the most UV-C tolerant isolate, although similar pigmentation was detected as compared to most of the strains investigated. Catalase activity in this strain was one of the strongest. Interestingly, ATCC 6538 *S. aureus*, which was shown to have both the highest catalase activity and the strongest pigmentation, was not more tolerant than the other isolates.

The fact that highly pigmented *S. aureus* isolates do not appear to be associated with greater UV-C tolerance is consistent with the study of Pannu et al. [19]. On the one hand, they showed that *crt* mutants of *S. aureus* lacking staphyloxanthin expression have lower UV-C tolerance compared to wild-type. On the other hand, overexpression of staphyloxanthin in the complemented mutant *S. aureus* is associated with decreased UV-C tolerance compared to wild-type. They suggested that overexpression leads to increased absorption, which in turn contributes to more severe oxidative stress damage [19].

It is described that catalase-producing microorganisms are possibly able to reduce the efficacy of vaporized hydrogen peroxide [22]. The mechanism for this is based on oxidative reactions which among others is also a mechanism of UV-C radiation. However, in the case of short-term high effective UV-C irradiation, this effect was not observed. Moreover, no association of the individual capacity to express a biofilm-positive phenotype with susceptibility or tolerance against UV-C radiation could be observed in the investigated strains.

The different expression of several virulence factors of genetically highly diverse wild-type *S. aureus* had no influence on susceptibility to UV-C. In the present study, no targeted deletion mutants lacking pigmentation, catalase activity and biofilm formation, nor respective complemented mutants were examined. Hence, no statement can be made about whether the presence of such virulence factors has a confirmed influence on UV-C susceptibility. It should also be noted that it cannot be ruled out that pigment expression during growth on rich media in vitro may differ from the expression on a hospital surface to be disinfected. However, the observed diversity of pigmentation in wild-type strains does not have any influence. In addition, it must be taken into account that other clonal variations such as switching to small colony variants or spontaneous deletions in global regulators could have an effect on UV-C tolerance [19,45,46]. When assessing the results of the quantification of biofilm formation, it must be taken into account that only two stress factors potentially influencing biofilm formation were investigated. In addition, this in vitro test differs from the real conditions on site where, among other things, other temperatures and mechanical effects on surfaces can have an influence. When testing the UV-C tolerance of the different clinical isolates, resuspended cells were applied to surfaces and dried, without the steps of primary attachment and subsequent cell-to-cell adhesion to form a multi-layered biofilm. Therefore, we suggest no relevant amounts of PIA/PNAG for most strains. However, in suspensions of MRSA ST45/t015, individual larger clusters might have been present as this strain displays a strong biofilm-positive phenotype under all tested in vitro conditions. It is therefore not possible to claim whether the differences between the clinical isolates in biofilm formation lead to different UV-C tolerances. Since it has been shown that dry biofilms can contain relevant bacteria [27] and observations have also been published that bacterial biofilms are more tolerant than planktonic bacteria to disinfectants and environmental stress [25] but also to UV-C [25,26], further work on the influence of dry biofilms on inanimate surfaces on the disinfection success with UV-C would be interesting.

## 5. Conclusions

To conclude, observed phenotypic differences in catalase formation, pigmentation and biofilm formation in various clinical *S. aureus* isolates did not affect efficacy using highly effective short-term UV-C radiation. This is of relevance for the set-up and efficacy assessment of UV-C whole-room disinfection procedures in hospital environments. Moreover, results of frequently used reference strains also seem to be representative for clinical isolates in this species.

## Figures and Tables

**Figure 1 microorganisms-11-01332-f001:**
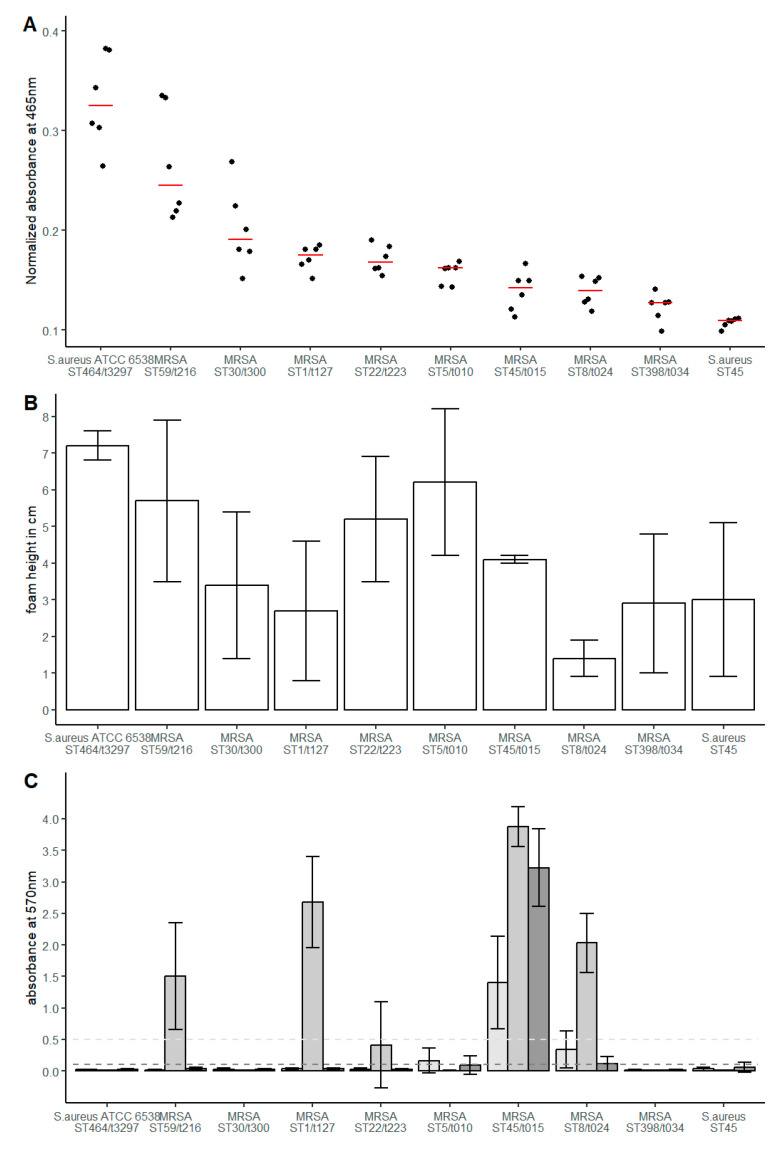
Quantification of pigmentation (**A**) catalase level (**B**) and biofilm formation (**C**) in different clinical *S. aureus* isolates. Staphyloxanthin levels (**A**) were determined by absorbance 465 nm and were normalized to the OD_600_ of low-pigmented ST45. The dots represent the values (*n* = 6) determined in three independent experiments. The median value of each isolate is shown by red horizontal lines. Catalase activity (**B**) is illustrated by foam height determined in five independent experiments. The median value is represented by bar height, which is supplemented by error bars showing the interquartile range. Biofilm formation (**C**) was evaluated in three independent experiments, resulting in 48 values per condition and strain. The bars show the mean value of the A_570_ value, supplemented by the standard deviation, which is represented as an error bar. The colouring of the bars represents the growth medium used (TSB: light grey; TSB + 4% NaCl: medium grey; TSB + 4% ethanol: dark grey). A_570_ values < 0.1 were defined as biofilm-negative (dark grey dotted line) whereas values > 0.5 corresponded strong biofilm formation (light grey dotted line), respectively.

**Figure 2 microorganisms-11-01332-f002:**
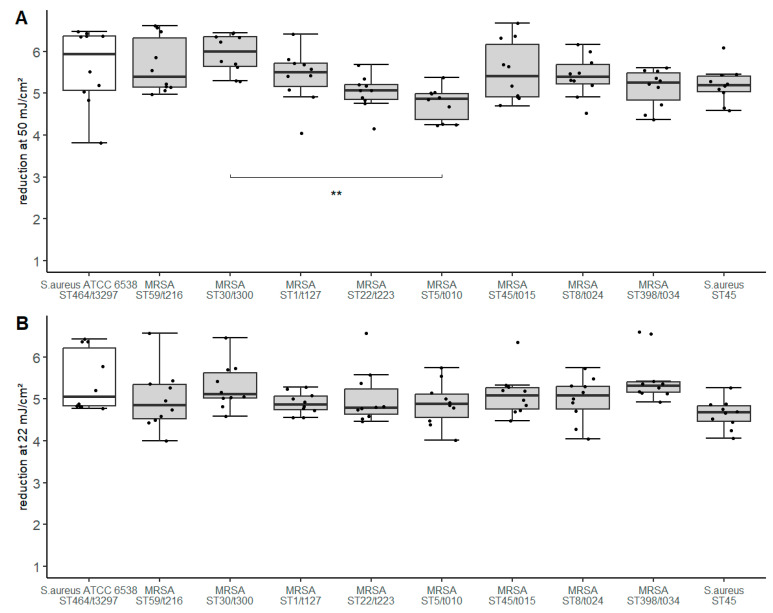
Log reduction values (LRV) of clinical *S. aureus* isolates after irradiation with a UV-C dose of 50 mJ/cm^2^ (**A**) and 22 mJ/cm^2^ (**B**). Boxplots represent LRV determined in five independent experiments. Scatter of the individual measured values is represented by the individual points. The unfilled boxplot display the standard test organism. Significant result (*p* < 0.001 = **) of statistical pairwise comparison in (**A**) is shown by the bracket.

## Data Availability

The data presented in this study are available in an insert article.

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
