# Peer review of "Phenotypic Variation in Clinical S. aureus Isolates Did Not Affect Disinfection Efficacy Using Short-Term UV-C Radiation"

_microorganisms, 2023, doi:10.3390/microorganisms11051332_

Round 1
Reviewer 1 Report
Comments to authors:
-The current study is interesting; however, the authors should address the following comments to improve the quality of the manuscript:
-The manuscript should be revised for English editing and grammar mistakes.
- Please write the scientific names of bacterial pathogens and genes in the correct form all over the manuscript and the references section.
Title:
I think the work would benefit from the title that contains the main conclusion of the study (should be derived from the conclusion). Please modify the title.
Abstract:
- The abstract must illustrate the used methods and the most prevalent results (give more hints about methods and results). Besides, rephrase the aim of the work and the main conclusion of your findings.
-A graphical abstract is recommended (If possible).
- Add the full expression before the abbreviations.
-Introduction: (it needs to be more informative):
-Give a hint about the virulence factors and the mechanism of disease occurrence, and infecions caused by S. aureus.
- The authors should illustrate the public health importance concerning the emergence of multidrug-resistant (MDR) bacterial pathogens that reflect the necessity of new potent and safe antimicrobial agents. Several studies proved the widespread MDR- bacterial pathogens;
Authors could add the following paragraph:
Multidrug resistance has been increased all over the world that is considered a public health threat. Several recent investigations reported the emergence of multidrug-resistant bacterial pathogens from different origins that increase the necessity of the proper use of antibiotics. Besides, the routine application of the antimicrobial susceptibility testing to detect the antibiotic of choice as well as the screening of the emerging MDR strains. You are advised to cite the following valuable studies:
1. PMID: 32397408
2. PMID: 36365013
3. PMID: 35971557
- Illustrate the mechanism of action different virulence factors of S. aureus.
- Give a hint about UV-C tolerance in S. aureus.
-Rephrase the aim of the work to be clear and better sound.
Material and methods:
- Support all methods with updated specific references.
• Add the company, city, and country of the used chemicals and reagents.
- Isolation and identification of S. aureus:
Discuss in detail the methods of isolation and identification of S.aureus. Besides, specific references should be added.
• Add the company, city, and country of the used bacterial media and reagents that were used in the biochemical identification of isolates. Also, enumerate all used biochemical reactions.
-Authors are advised to perform Antibiotic susceptibility testing:
-Please, explain in detail
•Add the names of the antimicrobial classes and enumerate the tested antibiotics.
•The authors are advised to classify the tested isolates to MDR , XDR, and PDR as described by Magiorakos et al.
Magiorakos AP, Srinivasan A, Carey RB, Carmeli Y, Falagas ME, Giske CG, et al. Multidrug-resistant, extensively drug-resistant and pandrug-resistant bacteria: An international expert proposal for interim standard definitions for acquired resistance. Clin Microbiol Infect. 2012; 18:268–81. doi:10.1111/j.1469-0691.2011.03570.x.
- The detection of virulence and antimicrobial resistance genes in the recovered isolates should be performed. Afterwards, the correlation between phenotypic and genotypic multidrug resistance should be performed (If available, or add this point to the study limitations).
-Give more details about the software used in statistical analyses.
-Results:Good presentation:
- Please add a starting paragraph to the results section to briefly introduce the topic, your goals and
hypothesis and a short summary of what you did in this work.
-Add this subtitle: Phenotypic characteristics of the recovered isolates:
• Illustrate in detail the phenotypic characteristics of the recovered S. aureus
isolates.
-Antimicrobial susceptibility testing:
• -Illustrate in a new table the occurrence of MDR (Multidrug resistance) among the recovered isolates as the following (illustrate the names of the antimicrobial classes and different antibiotics):
No. of strains % Type of resistance
R, MDR, and XDR Phenotypic multidrug resistance
(Antimicrobial classes and different antibiotics). The antibiotic-resistance genes
-The correlation (Correlation coefficient) between phenotypic and genotypic multidrug resistance should be performed. (If available).
-Illustrate the correlation between biofilm formation and antimiceobial resistance.
-Increase the resolution of all figures (must be 600 dpi).
-Discussion:
The authors are advised to illustrate the real impact of their findings without repetition of results.
- Please illustrate the mechanism of UV-C tolerance in S. aureus.
- Please illustrate different mechanisms of antimicrobial resistance in S. aureus.
-Conclusion
- Should be rephrased to be sounded. A real conclusion should focus on the question or claim you articulated in your study, which resolution has been the main objective of your paper?
The manuscript should be revised for English editing and grammar mistakes.
- Please write the scientific names of bacterial pathogens and genes in the correct form all over the manuscript and the references section.
Author Response
Thank you very much for the prompt and constructive review. We have implemented almost all suggestions completely. A detailed overview of the changes and, if necessary, a justification for slight deviations from the recommendations can be found in the attached file.
All infection prevention measures, such as final disinfection, which was the main subject of this study, are recommended for all MRSA regardless of additional resistance mechanisms. Therefore, we did not address this aspect at all suggested parts of the manuscript. For interested readers, detailed information on resistance is now available in the supplementary material.

Reviewer 2 Report
In this manuscript, authors described the effectiveness of short-term UV-C radiation against wide spectrum of Staphylococcus aureus strains irrespective of phenotypic variations (i.e., catalase formation, pigmentation, and biofilm formation) by analyzing the nine genetically different clinical isolates and one reference strain S. aureus ATCC 6538.
Well-written manuscript with scientifically sound methods and analysis.
This reviewer noticed some minor points as follows:
1. How did authors select HA-MRSA, CA-MRSA and LA-MRSA strains included in this study? It should be stated in materials and methods section.
2. Result section, line 215, 223 and 241, ST45 should be specified as MSSA or MRSA.
3. S. aureus should be italicized throughout the manuscript.
Author Response
Thank you very much for the prompt and constructive review. We have implemented all your suggestions. A detailed overview of the changes can be found in the attached file. Based on the manifold suggestions of the further reviewer, the manuscript was fundamentally changed in parts and corrected again by a native speaker.

Round 2
Reviewer 1 Report
The authors have carried out significant changes to the manuscript. They have addressed most of the suggested corrections and comments. Really, it's an interesting study that has a significant impact. Now, the manuscript could be accepted.
Congratulations.